# Association between Smoking Status and Incident Non-Cystic Fibrosis Bronchiectasis in Young Adults: A Nationwide Population-Based Study

**DOI:** 10.3390/jpm12050691

**Published:** 2022-04-26

**Authors:** Bumhee Yang, Kyungdo Han, Bongseong Kim, Hyung Koo Kang, Jung Soo Kim, Eung-Gook Kim, Hayoung Choi, Hyun Lee

**Affiliations:** 1Division of Pulmonary and Critical Care Medicine, Department of Internal Medicine, Chungbuk National University Hospital, Chungbuk National University College of Medicine, Cheongju 28644, Korea; ybhworld0415@gmail.com; 2Department of Statistics and Actuarial Science, Soongsil University, Seoul 06978, Korea; hkd917@naver.com (K.H.); qhdtjd12@gmail.com (B.K.); 3Division of Pulmonary and Critical Care Medicine, Department of Internal Medicine, Ilsan Paik Hospital, Inje University College of Medicine, Goyang 10380, Korea; inspirit26@gmail.com; 4Division of Pulmonary and Critical Care Medicine, Department of Internal Medicine, Inha University College of Medicine, Incheon 22332, Korea; acecloer31@gmail.com; 5Medical Research Center, Department of Biochemistry, Chungbuk National University College of Medicine, Cheongju 28644, Korea; egkim@chungbuk.ac.kr; 6Division of Pulmonary, Allergy, and Critical Care Medicine, Department of Internal Medicine, Hallym University Kangnam Sacred Heart Hospital, Hallym University College of Medicine, Seoul 07440, Korea; 7Division of Pulmonary Medicine and Allergy, Department of Internal Medicine, Hanyang University College of Medicine, Seoul 04763, Korea

**Keywords:** bronchiectasis, smoking, risk factor, epidemiology

## Abstract

Smoking traditionally has not been considered as a cause of bronchiectasis. However, few studies have evaluated the association between smoking and bronchiectasis. This study aimed to investigate the association between smoking status and bronchiectasis development in young adults. This study included 6,861,282 adults aged 20–39 years from the Korean National Health Insurance Service database 2009–2012 who were followed-up until the date of development of bronchiectasis, death, or 31 December 2018. We evaluated the incidence of bronchiectasis according to smoking status. During a mean of 7.4 years of follow-up, 23,609 (0.3%) participants developed bronchiectasis. In multivariable Cox regression analysis, ex-smokers (adjusted hazard ratio (aHR) = 1.07, 95% confidence interval (CI) = 1.03–1.13) and current-smokers (aHR = 1.06, 95% CI = 1.02–1.10) were associated with incident bronchiectasis, with the highest HR in ≥ 10 pack-years current smokers (aHR = 1.12, 95% CI = 1.06–1.16). The association of smoking with bronchiectasis was more profound in females than in males (*p* for interaction < 0.001), in younger than in older participants (*p* for interaction = 0.036), and in the overweight and obese than in the normal weight or underweight (*p* for interaction = 0.023). In conclusion, our study shows that smoking is associated with incident bronchiectasis in young adults. The association of smoking with bronchiectasis development was stronger in females, 20–29 year-olds, and the overweight and obese than in males, 30–40-year-olds, and the normal weight or underweight, respectively.

## 1. Introduction

Non-cystic fibrosis bronchiectasis (hereafter referred to as bronchiectasis), a chronic lung disease characterized by irreversible dilatation and destruction of airways, has been considered an orphan disease [1]. However, in terms of medical cost and mortality, the prevalence and disease burden of bronchiectasis have been increasing worldwide [1,2,3,4], indicating that this disease is an orphan no longer [5,6]. Thus, global strategies are needed to reduce the disease burden of bronchiectasis, which requires exploration of associated factors of bronchiectasis.

Current guidelines for diagnosis of bronchiectasis recommend comprehensive work-up to evaluate the etiology of bronchiectasis [7,8], which includes asthma, chronic obstructive pulmonary disease (COPD), immunodeficiencies, autoimmune diseases, alpha one antitrypsin deficiency, primary ciliary dyskinesia, and previous respiratory infections. Despite these workups, many cases of bronchiectasis are idiopathic [9], suggesting the presence of risk factors associated with personal habits and social, regional, and environmental factors. Smoking is one of the most important factors affecting the development of lung diseases such as COPD, lung cancer, and pulmonary tuberculosis (TB) [10,11,12,13]. Regarding bronchiectasis and smoking, several registry studies have suggested smoke exposure as a risk factor in pediatric patients with bronchiectasis [14,15,16]. However, it is largely unknown whether smoking is related to development of bronchiectasis in adults. Considering the importance of prevention of chronic disease at an earlier stage, the association of smoking with bronchiectasis in young adults could provide clinically relevant information.

Hence, this study aimed to investigate the association of smoking status with development of bronchiectasis in young adults using a nation-wide cohort of young adults.

## 2. Methods

### 2.1. Study Population

We used the Korean National Health Insurance Service (NHIS) database, the single-payer universal health system of Republic of Korea. The NHIS maintains claims data on all reimbursed inpatient and outpatient visits, procedures, and prescriptions. Additionally, the NHIS database includes data from annual or biennial health screening exams provided free of charge by the Ministry of Health and Welfare. Regardless of age, employees that pay insurance premiums, including the self-employed, are eligible for health examination every two years; employees engaged in manual labour receive health examination every year. For those who do not pay insurance premiums, health examination is provided to individuals ≥ 40 years of age every two years. Approximately 70–80% of all eligible persons underwent screening [17].

This study initially included 6,861,282 individuals, aged 20–39 years, who participated in a health screening exam between 1 January 2009 and 31 December 2012. We excluded the participants who had missing information for at least one of variables analyzed in this study (n = 543,778), those diagnosed with bronchiectasis before the enrollment period (n = 35,005), those diagnosed with cystic fibrosis, situs inversus, or congenital bronchiectasis before the enrolment period (n = 376), those diagnosed with bronchiectasis within one year after enrollment (n = 6560), and those who died within one year after enrollment (n = 6498). Finally, a total of 6,275,575 participants were included in this study (Figure 1). The cohort was followed from the enrollment date to the date of bronchiectasis development, death, or the end of the study period (31 December 2018), whichever came first (Appendix A).

The study protocol was approved by the Institutional Review Board of Chungbuk National University Hospital (No. 2022-01-009). The requirement for informed consent was waived because the NHIS database was constructed after anonymization of patient data.

### 2.2. Exposure: Smoking Status

We obtained information on smoking status from the health examination database. Participants were asked to complete a self-administered questionnaire and to provide categorical responses to questions on smoking status (i.e., never smoker, ex-smoker, or current smoker). Current smokers recorded the total duration of smoking (years) and the average number of cigarettes smoked daily. The cumulative smoking exposure was reported as pack-years (PY) by multiplying the average cigarette consumption per day (packs) and the smoking period (years). We divided current smokers into two groups according to cumulative smoking exposure: less than 10 PY and more than 10 PY.

### 2.3. Outcome: Incident Bronchiectasis

The main study outcome was incidence of bronchiectasis according to smoking status. Bronchiectasis was defined by the following criteria: (1) at least one claim under the International Statistical Classification of Diseases and Related Health Problems, 10th revision (ICD-10) diagnosis code J47, and (2) exclusion of individuals with cystic fibrosis, situs inversus or congenital bronchiectasis [18,19,20,21].

### 2.4. Covariates

Covariates were determined based on the literature and data availability and included socioeconomic status, lifestyle factors, and comorbidities. Body mass index (BMI) was calculated as weight in kilograms divided by square of height in meters and was categorized according to Asian-specific criteria [22]. Data on alcohol consumption and regular physical activity were collected from self-administered questionnaires. Categories for alcohol consumption were none (0 g/day), mild (<30 g/day), and heavy (≥30 g/day) [19,23,24]. Categories for exercise were regular (>30 min of moderate physical at least 5 times per week or >20 min of strenuous physical activity at least 3 times per week) and non-regular [19,23,24]. Income level was dichotomized at the lowest 20%; low-income category also included Medicaid beneficiaries [19,23,24]. The residential area was stratified into urban and rural areas. Number of hospital visits counted the number of outpatient visits and the number of hospitalizations.

Comorbidities were assessed during the enrollment period and defined using the following ICD-10 codes: asthma (J45–J46), pulmonary TB (A15–A19), non-tuberculous mycobacterial infection (A310), diabetes mellitus (E11–E14), chronic kidney disease or end stage of renal disease (N18.1–N18.5 and N18.9), gastroesophageal reflux (K21), solid cancer (C00–C97), connective tissue disease (M05, M06, M32, M35, and M45), hematologic malignancy (C90, C910, C920, C921, C922, C924–926, C928, and C930), transplantation status (Z940, Z944, Z941, and Z942), human immunodeficiency virus (HIV) infection and acquired immune deficiency syndrome (AIDS) (B20–B24), and inflammatory bowel disease (K50–K51) [5,19,23,25].

### 2.5. Statistical Analysis

Baseline characteristics of participants are presented as means (standard deviations) or numbers (%) according to smoking status. The incidence rate of bronchiectasis was calculated by dividing the number of events by 1000 person-years (PY). Cox proportional hazards regression analyses were conducted to obtain the hazard ratios (HRs) and 95% confidence intervals (CIs) for the occurrence of bronchiectasis based on smoking status. Model 1 was unadjusted. In multivariable analyses, age, sex, BMI, alcohol consumption, regular exercise, low income, area of residence (rural or urban), and the number of hospital visits were adjusted in Model 2; comorbidities that might be associated with the occurrence of bronchiectasis (respiratory disease, connective tissue disease, solid cancer, hematologic malignancy, transplantation status, immunodeficiency, inflammatory bowel disease, and HIV and AIDS) were further adjusted in Model 3. Statistical analyses were conducted using SAS software (version 9.4; SAS Institute, Cary, NC, USA), and statistical significance was defined as a two-sided *p* value < 0.05.

## 3. Results

### 3.1. Baseline Characteristics

Table 1 summarized the baseline characteristics of the study population according to smoking status. The never-smokers were the youngest (30.1 ± 5.1 years), had the lowest BMI (22.2 ± 3.5 kg/m^2^), and had the smallest proportion of regular exercisers (11.3%) among the groups (*p* < 0.001 for all). However, the proportion of urban residents (48.9%) was highest in never-smokers (*p* < 0.001). On the other hand, current smokers had the highest proportions of males (93.2%) and of heavy alcohol consumption (17.3%) across smoking status categories (*p* < 0.001 for both).

Regarding comorbidities, current smokers had the largest proportion of diabetes mellitus (2.9%), whereas never smokers had the largest proportions of gastroesophageal reflux (11.9%) and respiratory disease (6.1%), most of which was asthma (6.0%), compared with other smoking status categories (*p* < 0.001 for all).

### 3.2. The Incidence of Bronchiectasis and Smoking

During a mean follow-up period of 7.4 (±1.1) years, 23,609 (0.3%) participants developed bronchiectasis. As shown in Figure 2, the cumulative incidence of bronchiectasis was significantly different according to smoking status (log-rank test, *p* < 0.001). As shown in Table 2, the HR of bronchiectasis development was significantly increased in ex-smokers (adjusted hazard ratio (aHR) in the Model 3 = 1.07, 95% CI 1.03–1.13) and current-smokers (aHR = 1.06 in the Model 3, 95% CI = 1.02–1.10), with the highest HR in ≥10 PY smokers (aHR in the Model 3 = 1.12, 95% CI = 1.06–1.16). However, the HR was not increased in <10 PY current smokers when compared to never-smokers (aHR in the Model 3 = 1.03, 95% CI = 0.98–1.07).

The association of smoking with incident bronchiectasis was more profound in females than in males (*p* for interaction < 0.001), in the younger than in the older (*p* for interaction = 0.036), and in the overweight and obese than in the normal weight or underweight (*p* for interaction = 0.023).

### 3.3. Effect of Comorbid Profiles on the Relationship between Smoking Status and Incident Bronchiectasis

We further investigated the impact of comorbid profiles on the association of bronchiectasis with smoking status, which are related to the development of bronchiectasis (Table 3). Comorbid profiles (respiratory diseases, connective tissue diseases, inflammatory bowel diseases, and immunocompromised diseases) did not have a significant impact on the association between smoking status and incident bronchiectasis (*p* for interaction > 0.05 for all comorbidities).

## 4. Discussion

This population-based longitudinal cohort study assessed the association of smoking status with incident bronchiectasis among young adults. Smokers were more likely to be diagnosed with bronchiectasis than never-smokers, which was especially high in ≥10 PY current smokers. The association of smoking with bronchiectasis was more profound in females than in males, in the younger (20–29 years) than in the older (30–39 years), and in the overweight and obese than in the normal weight or underweight. On the other hand, comorbid profiles (respiratory diseases, connective tissue diseases, inflammatory bowel diseases, and immunocompromised diseases) did not have a significant impact on the association between smoking status and bronchiectasis.

Smoking is one of the most important factors influencing the development of chronic lung diseases such as COPD, lung cancer, and pulmonary TB [10,11,12,13]. However, the association between smoking and incident bronchiectasis has not been well elucidated. Studies so far have focused only on the association between smoking and the worse clinical outcomes of patients with bronchiectasis [25,26]. To the best of our knowledge, this study is the first to evaluate the association between smoking status and incident bronchiectasis in young adults. Our study also has the advantage of considering personal habits, socioeconomic factors, and comorbidities that might affect the development of bronchiectasis.

One possible explanation for the higher association with bronchiectasis in smokers when compared to never-smokers might be related to smoking-related impaired lung defense mechanisms, which increase the risk of respiratory infections. Chronic infection is a major component of the pathophysiologic mechanism of bronchiectasis [27]. Smoking facilitates recurrent respiratory infection through alterations in mechanisms of the host defense system including structural changes in the respiratory tract, a decrease in mucociliary clearance, and a decrease in immune response such as decreased secretary immunoglobulin [28,29,30]. Furthermore, it is also well demonstrated that respiratory infection can be a key player in the development of bronchiectasis in patients with COPD [28]. Another possible explanation is that smoking increases the inflammatory activities of the airways. Oxides in cigarette smoke can cause airway inflammation and tissue damage through various kinds of protease including neutrophil elastase and matrix metalloproteinase [31], which are known to play a critical role in the development and progression of bronchiectasis. For example, neutrophil elastase destroys the extracellular matrix, increases mucous gland proliferation and mucus production, reduces ciliary body beat rate, and damages airway epithelium directly [32,33,34]. Furthermore, the effect of smoking on incident bronchiectasis was significant in ≥10 PY smokers as compared to never smokers, although the effect was insignificant in <10 PY smokers. The phenomenon might be explained by the dose–response relationships; smoking amounts of <10 PY might not be enough to result in incident bronchiectasis.

Interestingly, our subgroup analyses showed that the association of smoking history with incident bronchiectasis was greater in females, the younger (20–29 years), and the overweight and obese than in males, the older (30–39 years), and the normal weight or underweight, respectively. The risk of developing a chronic respiratory disease associated with smoking, such as lung cancer and COPD, is greater in females than in males, and it is thought to be related to sex disparity [35]. Supporting this, it was found that smoking affects the development of COPD and lung cancer by increasing the expression of the estrogen receptor. Smoking aggravates the effects of estrogen and endocrine disruptive chemicals that target the estrogen receptor to contribute to lung carcinogenesis [36]. In addition, a previous study showed that the expression of estrogen receptors in lung tissue in a mouse model of COPD was associated with increased small airway remodeling in females when compared to male mice with chronic smoke exposure [37]. The excess risk of small airway disease in female mice after chronic smoke exposure was associated with increased oxidative stress and TGF-β1 signaling, and estrogen might regulate this oxidant/TGF-β signaling axis. Although little information is available about the mechanism of how smoking influences the development of bronchiectasis, these previous findings suggest that similar pathophysiology might be applicable in bronchiectasis [37,38].

Obesity can play a significant role in the pathogenesis of pulmonary diseases through mechanisms that involve the release of proinflammatory mediators by adipose tissue [39]. Inflammatory responses in the lungs are influenced by adipocytokines, leptin and adiponectin, cytokines, acute phase proteins, and other mediators produced by adipose tissue [40,41,42]. These findings support the notion that obesity increases lung inflammation. Smoking has been reported to exacerbate systemic inflammation caused by obesity and is believed to increase lung inflammation. Because of the free radicals present in smoking, and the inflammatory response they induce, obesity-induced oxidative stress (production of reactive oxygen species) is enhanced, which increases systemic inflammation [43]. These mechanisms suggest that smoking has a greater effect on the obese than on those of normal weight or underweight in the development of bronchiectasis. Our study also showed that the effect of smoking on the development of bronchiectasis is greater with younger than older. This finding renders an important clinical implication emphasizing that quitting smoking might be more effective in younger participants than in older participants to prevent bronchiectasis. However, since the mechanism explaining this phenomenon is not clear, thus, future studies are needed on this issue.

There are several limitations to this study. First, the diagnosis of bronchiectasis was based on the ICD-10 diagnosis code and not on chest computed tomography results. Thus, there could be misclassification of the diagnosis of bronchiectasis. Second, there might be a bias that smokers underwent more chest computed tomography than did non-smokers, which might have led to more diagnoses of bronchiectasis in smokers than in never-smokers. Third, due to the lack of data on childhood respiratory infection, an important risk factor of bronchiectasis in young adults, our study could not consider this in our analyses. Fourth, as we solely focused on the smoking amount and the risk of bronchiectasis in current smokers; the association between smoking amount and the risk of bronchiectasis in ex-smokers should be investigated in future studies.

## 5. Conclusions

This large, nationwide, longitudinal study demonstrated that smoking history is associated with incident bronchiectasis development in young adults. The association of smoking with incident bronchiectasis was more prominent in females, younger individuals (20–29 years), and overweight and obese participants than in their respective counterparts.

## Figures and Tables

**Figure 1 jpm-12-00691-f001:**
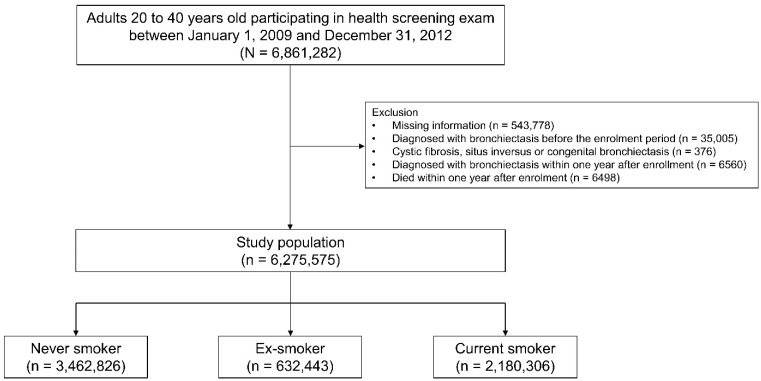
Flow chart of the study population.

**Figure 2 jpm-12-00691-f002:**
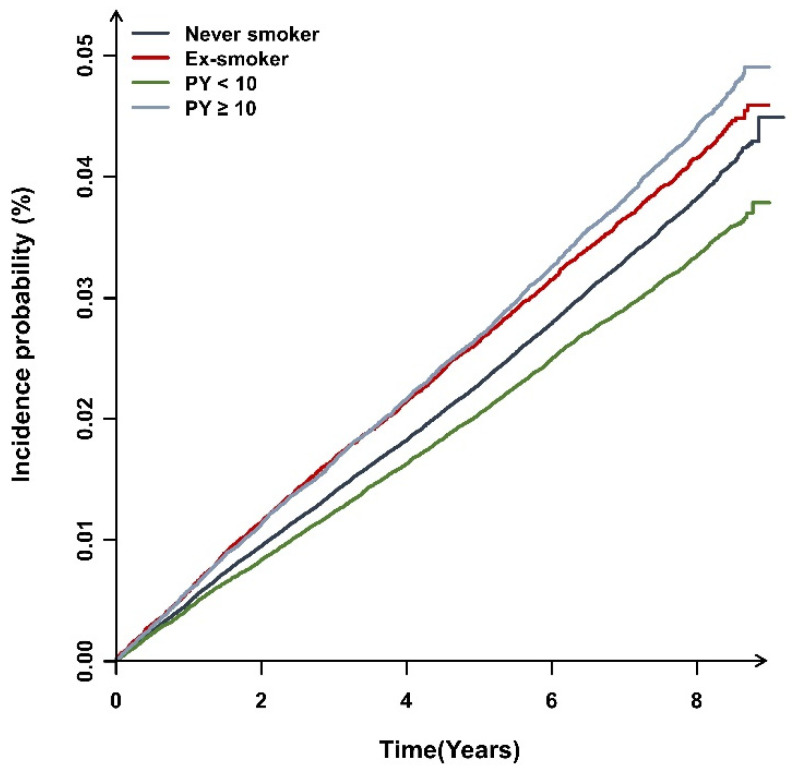
Cumulative incidence of bronchiectasis according to smoking status. Abbreviation: PY, pack-years.

**Table 1 jpm-12-00691-t001:** Baseline characteristics of the study population according to smoking status.

		Smoking Status			
	Total (N = 6,275,575)	Never Smoker (n = 3,462,826)	Ex-Smoker (n = 632,443)	Current Smoker (n = 2,180,306)	*p*-Value
**Age, years**	30.8 ± 4.9	30.1 ± 5.1	32.5 ± 4.64	31.4 ± 4.74	<0.001
<30	2,655,026 (42.4)	1,675,479 (48.4)	175,548 (27.7)	803,999 (36.8)	<0.001
≥30	3,620,549 (57.6)	1,787,347 (51.6)	456,895 (72.3)	1,376,307 (63.2)	
**Sex**					<0.001
Male	3,712,379 (59.2)	1,135,390 (32.8)	546,321 (86.4)	2,030,668 (93.2)	
Female	2,563,196 (40.8)	2,327,436 (67.2)	86,122 (13.6)	149,638 (6.8)	
**BMI (kg/m^2^)**	23.0 ± 3.6	22.2 ± 3.5	24.0 ± 3.3	24.0 ± 3.6	<0.001
<18.5 kg/m^2^	476,686 (7.6)	371,396 (10.7)	19,801 (3.1)	85,489 (3.9)	<0.001
18.5–22.9 kg/m^2^	2,931,905 (46.7)	1,885,080 (54.4)	227,953 (36.0)	818,872 (37.5)	
23–24.9 kg/m^2^	1,203,883 (19.2)	557,097 (16.1)	158,130 (25.0)	488,656 (22.4)	
≥25 kg/m^2^	1,663,101 (26.5)	649,253 (18.8)	226,559 (35.8)	787,289 (36.1)	
**Alcohol consumption**					<0.001
None	2,367,631 (37.7)	1,811,004 (52.3)	141,565 (22.4)	415,062 (19.1)	
Mild	3,353,737 (53.5)	1,557,776 (45.0)	408,841 (64.6)	1,387,120 (63.6)	
Heavy	554,207 (8.8)	94,046(2.7)	82,037 (13.0)	378,124 (17.3)	
**Regular exercise**					<0.001
No	5,470,001 (87.2)	3,069,862 (88.7)	518,788 (82.0)	1,881,351 (86.3)	
Yes	805,574 (12.8)	392,964 (11.3)	113,655 (18.0)	298,955 (13.7)	
**Low income**					<0.001
No	5,276,741 (84.1)	2,812,455 (81.2)	567,943 (89.8)	1,896,343 (87.0)	
Yes	998,834 (15.9)	650,371 (18.8)	64,500 (10.2)	283,963 (13.0)	
**Residence**					<0.001
Rural	3,276,315 (52.2)	1,767,983 (51.1)	330,394 (52.2)	1,177,938 (54.0)	
Urban	2,999,260 (47.8)	1,694,843 (48.9)	302,049 (47.8)	1,002,368 (46.0)	
**Number of hospital visits**	3.6 ± 6.5	3.9 ± 6.9	4.2 ± 7.2	2.9 ± 5.7	<0.001
Admission	0.1 ± 0.3	0.1 ± 0.3	0.1 ± 0.3	0.1 ± 0.3	<0.001
Outpatient	3.6 ± 6.5	3.9 ± 6.8	4.1 ± 7.1	2.9 ± 5.6	<0.001
**Comorbidities**					
DM	122,007 (1.9)	44,005 (1.2)	15,258 (2.4)	62,744 (2.9)	<0.001
CKD or ESRD	2317 (0.1)	1280 (0.1)	483 (0.1)	554 (0.1)	<0.001
GERD	674,566 (10.8)	414,329 (11.9)	74,335 (11.7)	185,902 (8.5)	<0.001
Respiratory disease	328,703 (5.3)	208,670 (6.1)	34,419 (5.5)	85,614 (4.0)	<0.001
Asthma	326,557 (5.2)	207,418 (6.0)	34,133 (5.4)	85,006 (3.9)	<0.001
TB	2418 (0.1)	1437 (0.1)	320 (0.1)	661 (0.1)	<0.001
NTM disease	104 (<0.1)	63 (<0.1)	17 (<0.1)	24 (<0.1)	0.013
Others					
Connective tissue disease	44,670 (0.7)	27,211 (0.8)	4878 (0.8)	12,581 (0.6)	<0.001
Solid cancer	12,563 (0.2)	9782 (0.3)	1685 (0.3)	1096 (0.1)	<0.001
Hematologic malignancy	14 (<0.1)	8 (<0.1)	6 (<0.1)	0 (<0.1)	<0.001
Transplantation status	53 (<0.1)	25 (<0.1)	19 (<0.1)	9 (<0.1)	<0.001
HIV/AIDS	441 (<0.1)	176 (<0.1)	60 (<0.1)	205 (<0.1)	<0.001
Immunodeficiency	295 (<0.1)	210 (<0.1)	26 (<0.1)	59 (<0.1)	<0.001
Inflammatory bowel disease	7906 (0.2)	4539 (0.1)	1275 (0.2)	2092 (0.1)	<0.001

Abbreviations: BMI, body mass index; DM, diabetes mellitus; CKD, chronic kidney disease; ESRD, end-stage renal disease; GERD, gastroesophageal reflux disease; TB, tuberculosis; NTM, nontuberculous mycobacteria; HIV, human immunodeficiency virus; AIDS, acquired immunodeficiency syndrome.

**Table 2 jpm-12-00691-t002:** Association of smoking status with the incidence of bronchiectasis.

						HR (95% CI)		
	Smoking Status	N	Bronchiectasis	Duration (PY)	IR per 1000	Model 1	Model 2	Model 3
**Overall**	Never smoker	3,462,826	12,884	25,333,8814	0.508	1 (reference)	1 (reference)	1 (reference)
	Ex-smoker	632,443	2639	4,731,442	0.557	1.09 (1.05,1.14)	1.08 (1.03,1.13)	1.07 (1.03,1.13)
	Current smoker	2,180,306	8086	16,133,463	0.501	0.98 (0.95,1.01)	1.05 (1.01,1.09)	1.06 (1.02,1.10)
	Pack years < 10	1,358,100	4453	9,963,491	0.446	0.87 (0.84,0.91)	1.03 (0.98,1.06)	1.03 (0.98,1.07)
	Pack years ≥ 10	822,206	3633	6,169,972	0.588	1.15 (1.11,1.19)	1.11 (1.05,1.15)	1.12 (1.06,1.16)
	*p* for trend					0.035	<0.001	<0.001
**Sex**								
Male	Never smoker	1,135,390	4029	8,376,524	0.480	1 (reference)	1 (reference)	1 (reference)
	Ex-smoker	546,321	2321	4,119,567	0.563	1.17 (1.11,1.23)	1.07 (1.01,1.12)	1.06 (1.01,1.12)
	Current smoker	2,030,668	7494	15,077,177	0.497	1.03 (0.99,1.07)	1.03 (0.99,1.07)	1.038(0.99,1.08)
	Pack years < 10	1,216,241	3911	8,960,985	0.436	0.90 (0.86,0.94)	0.98 (0.94,1.03)	0.99 (0.94,1.03)
	Pack years ≥ 10	814,427	3583	6,116,192	0.585	1.21 (1.16,1.27)	1.09 (1.04,1.14)	1.10 (1.05,1.15)
	*p* for trend					<0.001	0.018	0.005
Female	Never smoker	2,327,436	8855	16,957,357	0.522	1 (reference)	1 (reference)	1 (reference)
	Ex-smoker	86,122	318	611,874	0.519	0.99 (0.89,1.12)	1.07 (0.96,1.20)	1.06 (0.95,1.19)
	Current smoker	149,638	592	1,056,285	0.560	1.07 (0.99,1.17)	1.21 (1.11,1.31)	1.20 (1.10,1.31)
	Pack years < 10	141,859	542	1,002,505	0.540	1.04 (0.95,1.13)	1.18 (1.08,1.29)	1.18 (1.08,1.29)
	Pack years ≥ 10	7779	50	53,780	0.929	1.79 (1.36,2.37)	1.51 (1.14,2.00)	1.49 (1.12,1.97)
	*p* for trend					0.041	<0.001	<0.001
	*p* value					<0.001	<0.001	<0.001
	*p* for interaction					<0.001	<0.001	<0.001
**Age, years**								
<30	Never smoker	1,675,479	4484	12,126,911	0.369	1 (reference)	1 (reference)	1 (reference)
	Ex-smoker	175,548	505	1,276,722	0.395	1.06 (0.97,1.17)	1.13 (1.02,1.24)	1.12 (1.02,1.24)
	Current smoker	803,999	2127	5,826,907	0.365	0.98 (0.93,1.03)	1.10 (1.03,1.17)	1.10 (1.03,1.17)
	Pack years < 10	672,670	1730	4,855,497	0.356	0.96 (0.91,1.01)	1.09 (1.02,1.16)	1.09 (1.01,1.16)
	Pack years ≥ 10	131,329	397	971,410	0.408	1.10 (0.99,1.22)	1.18 (1.05,1.32)	1.18 (1.05,1.32)
	*p* for trend					0.904	0.001	0.001
≥30	Never smoker	1,787,347	8400	13,206,970	0.636	1 (reference)	1 (reference)	1 (reference)
	Ex-smoker	456,895	2134	3,454,719	0.617	0.97 (0.92,1.01)	1.06 (1.01,1.12)	1.05 (1.01,1.11)
	Current smoker	1,376,307	5959	10,306,556	0.578	0.91 (0.87,0.93)	1.03 (0.98,1.07)	1.03 (0.99,1.08)
	Pack years < 10	685,430	2723	5,107,993	0.533	0.83 (0.80,0.87)	0.98 (0.93,1.03)	0.98 (0.94,1.04)
	Pack years ≥ 10	690,877	3236	5,198,562	0.622	0.97 (0.93,1.01)	1.08 (1.03,1.14)	1.09 (1.04,1.15)
	*p* for trend					<0.001	0.029	0.001
	*p* for interaction					<0.001	0.022	0.036
**BMI (kg/m^2^)**								
<18.5	Never smoker	371,396	1716	2,709,061	0.633	1 (reference)	1 (reference)	1 (reference)
	Ex-smoker	19,801	111	144,234	0.769	1.21 (1.01,1.47)	0.98 (0.80,1.21)	0.98 (0.80,1.20)
	Current smoker	85,489	443	623,242	0.710	1.12 (1.01,1.24)	0.85 (0.74,0.98)	0.86 (0.75,0.99)
	Pack years < 10	64,695	292	468,357	0.623	0.98 (0.87,1.11)	0.86 (0.74,1.01)	0.86 (0.74,1.01)
	Pack years ≥ 10	20,794	151	154,884	0.974	1.53 (1.29,1.81)	0.85 (0.69,1.04)	0.85 (0.69,1.05)
	*p* for trend					0.001	0.038	0.046
18.5–22.9	Never smoker	1,885,080	7056	13,792,785	0.511	1 (reference)	1 (reference)	1 (reference)
	Ex-smoker	227,953	1080	1,698,419	0.635	1.24 (1.16,1.32)	1.16 (1.08,1.24)	1.15 (1.07,1.23)
	Current smoker	818,872	3209	6,054,509	0.530	1.03 (0.99,1.07)	1.04 (0.98,1.09)	1.04 (0.99,1.10)
	Pack years < 10	552,628	1917	4,050,839	0.473	0.92 (0.87,0.97)	1.02 (0.96,1.08)	1.02 (0.96,1.09)
	Pack years ≥ 10	266,244	1292	2,003,669	0.644	1.25 (1.18,1.33)	1.07 (1.01,1.15)	1.08 (1.01,1.16)
	*p* for trend					<0.001	0.163	0.099
23–24.9	Never smoker	557,097	1877	4,088,205	0.459	1 (reference)	1 (reference)	1 (reference)
	Ex-smoker	158,130	625	1,192,235	0.524	1.14 (1.04,1.24)	1.08 (0.98,1.19)	1.07 (0.97,1.19)
	Current smoker	488,656	1680	3,633,804	0.462	1.01 (0.94,1.07)	1.04 (0.96,1.12)	1.04 (0.96,1.13)
	Pack years < 10	298,815	887	2,203,629	0.402	0.87 (0.80,0.94)	0.99 (0.90,1.08)	0.99 (0.90,1.08)
	Pack years ≥ 10	189,841	793	1,430,175	0.554	1.20 (1.11,1.31)	1.11 (1.01,1.22)	1.12 (1.01,1.23)
	*p* for trend					0.064	0.151	0.110
≥25	Never smoker	649,253	2235	4,743,830	0.471	1 (reference)	1 (reference)	1 (reference)
	Ex-smoker	226,559	823	1,696,552	0.485	1.02 (0.94,1.11)	1.03 (0.94,1.12)	1.02 (0.94,1.12)
	Current smoker	787,289	2754	5,821,907	0.473	1.00 (0.94,1.06)	1.10 (1.03,1.17)	1.10 (1.03,1.18)
	Pack years < 10	441,962	1357	3,240,663	0.418	0.88 (0.83,0.95)	1.05 (0.97,1.13)	1.05 (0.97,1.13)
	Pack years ≥ 10	345,327	1397	2,581,243	0.541	1.14 (1.07,1.22)	1.16 (1.07,1.25)	1.17 (1.08,1.26)
	*p* for trend					0.046	<0.001	<0.001
	*p* for interaction					0.064	0.015	0.023

Model 1 was unadjusted; Model 2 was adjusted for age, sex, body mass index, alcohol consumption, regular exercise, low income, residential area, and number of hospital visits; Model 3 was additionally adjusted for respiratory disease, connective tissue disease, solid cancer, hematologic malignancy, transplantation status, immunodeficiency, inflammatory bowel disease, and HIV or AIDS. Abbreviations: HR, hazard ratio; CI, confidence interval; PY, person-years; IR, incidence rate; BMI, body mass index; HIV, human immunodeficiency virus; AIDS, acquired immunodeficiency syndrome.

**Table 3 jpm-12-00691-t003:** Impact of comorbidities on the relationship between smoking status and incident bronchiectasis.

						HR (95% CI)		
	Smoking Status	N	Bronchiectasis	Duration (PY)	IR per 1000	Model 1	Model 2	Model 3
**Respiratory disease**								
No	Never smoker	3,254,156	11,471	23,832,643	0.481	1 (reference)	1 (reference)	1 (reference)
	Ex-smoker	598,024	2380	4,477,618	0.531	1.10 (1.05,1.15)	1.08 (1.03,1.14)	1.08 (1.03,1.13)
	Current smoker	2,094,692	7490	15,510,580	0.482	1.02 (0.93,1.03)	1.06 (1.02,1.10)	1.06 (1.02,1.10)
	Pack years < 10	1,304,433	4127	9,577,112	0.430	0.89 (0.86,0.92)	1.03 (0.99,1.07)	1.03 (0.99,1.07)
	Pack years ≥ 10	790,259	3363	5,933,468	0.566	1.17 (1.13,1.22)	1.11 (1.06,1.17)	1.11 (1.06,1.17)
	*p* for trend					0.001	<0.001	<0.001
Yes	Never smoker	208,670	1413	1,501,238	0.941	1 (reference)	1 (reference)	1 (reference)
	Ex-smoker	34,419	259	253,823	1.020	1.08 (0.95,1.23)	1.01 (0.87,1.17)	1.01 (0.87,1.17)
	Current smoker	85,614	596	622,883	0.956	1.01 (0.92,1.12)	1.02 (0.91,1.15)	1.03 (0.93,1.14)
	Pack years < 10	53,667	326	386,379	0.843	0.89 (0.79,1.01)	0.98 (0.85,1.12)	0.98 (0.85,1.12)
	Pack years ≥ 10	31,947	270	236,504	1.141	1.21 (1.06,1.38)	1.09 (0.93,1.28)	1.09 (0.93,1.27)
	*p* for trend					0.197	0.435	0.448
	*p* for interaction					0.957	0.877	0.891
**Connective tissue disease**								
No	Never smoker	3,435,615	12,688	25,135,861	0.504	1 (reference)	1 (reference)	1 (reference)
	Ex-smoker	627,565	2597	4,695,583	0.553	1.09 (1.04,1.14)	1.07 (1.02,1.13)	1.07 (1.02,1.12)
	Current smoker	2,167,725	7995	16,041,918	0.498	0.98 (0.95,1.01)	1.05 (1.02,1.09)	1.05 (1.02,1.09)
	Pack years < 10	1,350,665	4411	9,909,679	0.445	0.88 (0.85,0.91)	1.02 (0.98,1.06)	1.02 (0.98,1.07)
	Pack years ≥ 10	817,060	3584	6,132,238	0.584	1.15 (1.11,1.19)	1.10 (1.05,1.15)	1.11 (1.06,1.16)
	*p* for trend					0.029	<0.001	<0.001
Yes	Never smoker	27,211	196	198,020	0.989	1 (reference)	1 (reference)	1 (reference)
	Ex-smoker	4878	42	35,858	1.171	1.18 (0.85,1.65)	1.35 (0.92,1.98)	1.34 (0.92,1.96)
	Current smoker	12,581	91	91,545	0.994	1.01 (0.78,1.29)	1.19 (0.86,1.64)	1.19 (0.86,1.63)
	Pack years < 10	7435	42	53,811	0.780	0.78 (0.56,1.10)	0.97 (0.66,1.42)	0.97 (0.66,1.42)
	Pack years ≥ 10	5146	49	37,733	1.298	1.31 (0.96,1.80)	1.57 (1.06,2.34)	1.58 (1.06,2.35)
	*p* for trend					0.483	0.114	0.111
	*p* for interaction					0.690	0.495	0.508
**Inflammatory bowel disease**								
No	Never smoker	3,458,287	12,862	25,300,827	0.508	1 (reference)	1 (reference)	1 (reference)
	Ex-smoker	631,168	2625	4,721,944	0.555	1.09 (1.04,1.13)	1.08 (1.03,1.13)	1.07 (1.02,1.12)
	Current smoker	2,178,214	8067	16,118,195	0.500	0.98 (0.95,1.01)	1.06 (1.01,1.09)	1.06 (1.02,1.09)
	Pack years < 10	1,356,726	4442	9,953,533	0.446	0.87 (0.84,0.90)	1.02 (0.98,1.06)	1.02 (0.98,1.06)
	Pack years ≥ 10	821,488	3625	6,164,661	0.588	1.15 (1.11,1.19)	1.10 (1.05,1.15)	1.11 (1.06,1.16)
	*p* for trend					0.042	<0.001	<0.001
Yes	Never smoker	4539	22	33,054	0.665	1 (reference)	1 (reference)	1 (reference)
	Ex-smoker	1275	14	9497	1.474	2.21 (1.13,4.33)	1.80 (0.86,3.78)	1.80 (0.85,3.78)
	Current smoker	2092	19	15,268	1.244	1.86 (1.01,3.45)	1.59 (0.79,3.19)	1.55 (0.77,3.11)
	Pack years < 10	1374	11	9957	1.104	1.65 (0.80,3.42)	1.46 (0.66,3.21)	1.42 (0.64,3.14)
	Pack years ≥ 10	718	8	5310	1.506	2.26 (1.01,5.08)	1.84 (0.74,4.55)	1.83 (0.74,4.54)
	*p* for trend					0.029	0.195	0.208
	*p* for interaction					0.110	0.138	0.149
**Solid cancer or Hematologic malignancy**								
No	Never smoker	3,453,036	12,830	25,263,418	0.507	1 (reference)	1 (reference)	1 (reference)
	Ex-smoker	630,754	2631	4,718,957	0.557	1.09 (1.05,1.14)	1.08 (1.03,1.13)	1.07 (1.02,1.12)
	Current smoker	2,179,210	8081	16,125,496	0.501	0.98 (0.95,1.01)	1.05 (1.02,1.09)	1.06 (1.02,1.09)
	Pack years < 10	1,357,444	4448	9,958,767	0.446	0.87 (0.85,0.91)	1.02 (0.98,1.06)	1.02 (0.98,1.06)
	Pack years ≥ 10	821,766	3633	6,166,728	0.589	1.15 (1.11,1.20)	1.10 (1.05,1.15)	1.11 (1.06,1.16)
	*p* for trend					0.027	<0.001	<0.001
Yes	Never smoker	9790	54	70,462	0.766	1 (reference)	1 (reference)	1 (reference)
	Ex-smoker	1689	8	12,484	0.640	0.82 (0.39,1.72)	1.26 (0.49,3.18)	1.26 (0.49,3.19)
	Current smoker	1096	5	7967	0.627	0.81 (0.32,2.02)	1.44 (0.48,4.28)	1.44 (0.48,4.28)
	Pack years < 10	656	5	4724	1.058	1.36 (0.54,3.41)	2.19 (0.77,6.23)	2.18 (0.76,6.20)
	Pack years ≥ 10	440	0	3243	0	NA	NA	NA
	*p* for trend					0.362	0.807	0.814
	*p* for interaction					0.637	0.657	0.651
**Immunocompromised disease ***								
No	Never smoker	3,462,415	12,880	25,330,891	0.508	1 (reference)	1 (reference)	1 (reference)z
	Ex-smoker	632,338	2638	4,730,674	0.557	1.09 (1.05,1.14)	1.08 (1.03,1.13)	1.07 (1.02,1.12)
	Current smoker	2,180,033	8085	16,131,533	0.501	0.98 (0.95,1.01)	1.05 (1.01,1.09)	1.06 (1.02,1.09)
	Pack years < 10	1,357,914	4453	9,962,181	0.446	0.87 (0.84,0.90)	1.02 (0.98,1.06)	1.02 (0.98,1.07)
	Pack years ≥ 10	822,119	3632	6,169,352	0.588	1.15 (1.11,1.19)	1.10 (1.05,1.15)	1.11 (1.06,1.16)
	*p* for trend					0.034	<0.001	<0.001
Yes	Never smoker	411	4	2989	1.337	1 (reference)	1 (reference)	1 (reference)
	Ex-smoker	105	1	767	1.303	0.98 (0.11,8.78)	2.24 (0.15,32.63)	1.50 (0.07,28.54)
	Current smoker	273	1	1929	0.518	0.38 (0.04,3.41)	1.31 (0.08,20.82)	2.18 (0.09,51.96)
	Pack years < 10	186	0	1309	0	NA	NA	NA
	Pack years ≥ 10	87	1	620	1.612	1.19 (0.13,10.72)	6.93 (0.24,199.33)	9.62 (0.22,416.78)
	*p* for trend					0.560	0.511	0.351
	*p* for interaction					0.992	0.987	0.863

Model 1 was unadjusted; Model 2 was adjusted for age, sex, BMI, alcohol consumption, regular exercise, low income, residential area, and number of hospital visits; Model 3 was additionally adjusted for respiratory disease, connective tissue disease, solid cancer, hematologic malignancy, transplantation status, immunodeficiency, inflammatory bowel disease, and HIV or AIDS. * Immunocompromised disease comprised immunodeficiency, organ transplantation, and HIV or AIDS. Abbreviations: HR, hazard ratio; CI, confidence interval; PY, person-years; IR, incidence rate; BMI, body mass index; HIV, human immunodeficiency virus; AIDS, acquired immunodeficiency syndrome; NA, not applicable.

## Data Availability

Data is available from the National Health Insurance Service database, the single-payer universal health system of Republic of Korea.

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
