# Peer review of "Association between Smoking Status and Incident Non-Cystic Fibrosis Bronchiectasis in Young Adults: A Nationwide Population-Based Study"

_jpm, 2022, doi:10.3390/jpm12050691_

Round 1

Reviewer 1 Report

Thank you for that interesting study evaluating smoking as risk factor for bronchictasis. The paper is well written and the topic of the work is new and significant. Although I do not believe that patient quit smoking due to these results. 

I have some minor comments:

l 67: please rewrite this sentence

l72: what does autimmune disease in relation to bronchiectasis mean? Please add other important etiologies (COPD, Asthma, AATD, 

ll175: Please specify etiology and comorbity. These are two different things.

Figure 1: Please discuss the lower incidence in patients with <10py in compare to never smoker.

ll287 this sentence is very similar 3 times in the manuscript. Please rewrite it.

Author Response

## Response to Reviewer 1

General comment.

Thank you for that interesting study evaluating smoking as risk factor for bronchictasis. The paper is well written and the topic of the work is new and significant. Although I do not believe that patient quit smoking due to these results. I have some minor comments.
Response. Thank you for your positive comments and kind appreciation. We have provided our point-by-point responses below for all your comments and suggestions.

Specific comments

Comment 1 (C1). Line 67: please rewrite this sentence

Response 1 (R1). Thank you for your comment. We have modified the sentence in the revised manuscript (page 4, lines 67–68) as follows:

“Thus, global strategies are needed to reduce the disease burden of bronchiectasis, which requires exploration of the risk factors of bronchiectasis.”

C2. Line 72: what does autimmune disease in relation to bronchiectasis mean? Please add other important etiologies (COPD, Asthma, AATD)

R2. Thank you for this pertinent suggestion. Accordingly, we have revised the sentence as follows (page 4, lines 70–72):

“…which includes asthma, chronic obstructive pulmonary disease (COPD), immunodeficiencies, autoimmune diseases such as alpha one antitrypsin deficiency…

C3. 175: Please specify etiology and comorbity. These are two different things.

R3. Thank you for raising this concern. Given the nature of population-based cohort studies, our dataset lacks information on the etiology of bronchiectasis. To address your concern, we have clarified the comorbidities in the Results section of the revised manuscript (page 9, line 177) as follows:

“Regarding comorbidities, current smokers…”

C4. Figure 1: Please discuss the lower incidence in patients with <10py in compare to never smoker.

R4. Thank you for bringing this to our attention. We apologize that this was not fully addressed in the original manuscript. The effect of smoking on incident bronchiectasis was significant in ≥10 PY smokers compared to never smokers, although the effect was insignificant in <10 PY smokers. This phenomenon could be explained by the dose-response relationships. Smoking amount <10 PY might not be enough to result in bronchiectasis development. We have discussed this issue in the revised manuscript (page 12, lines 239–243) as follows:

“Furthermore, the effect of smoking on incident bronchiectasis was significant in 10 PY smokers as compared to never smokers, although the effect was insignificant in <10 PY smokers. The phenomenon might be explained by the dose-response relationships; smoking amount <10 PY might not be enough to result in incident bronchiectasis.”

C5. 287 this sentence is very similar 3 times in the manuscript. Please rewrite it.

R5. We appreciate your careful review of our manuscript. Accordingly, we have revised the sentence as recommended (page 14, lines 295–297):

“The effect of smoking on bronchiectasis development was more prominent in females, younger individuals (20–29 years), and overweight and obese participants than in their respective counterparts.”

Reviewer 2 Report

The study by Yang and colleagues presents a large database analysis investigating the association between smoking and the development of bronchiectasis in young adults. The study is nicely performed, with limitations addressed. I have a few minor comments;

  1. It is stated that those diagnosed with bronchiectasis more than one year before the enrollment period were excluded. What about those diagnosed within a year of enrollment? If they were included, how were they handled in the survival analysis?
  2. IN model 3 comorbidities were adjusted for. Were all those diagnosed before the enrollment period? Were any diagnosed during follow-up, and if so, were they modelled as time-dependent covariates?
  3. The increased risk of bronchiectasis in ex-smokers is interesting. Were there any differences in this group based on their pack year history (greater or less than 10 pack years)? Was the risk seen with the higher pack-year exposure seen in current smokers also seen in the ex-smoker group? Was there any association with bronchiectasis and time since smoking ceased?

Author Response

## Response to Reviewer 2

General comment

The study by Yang and colleagues presents a large database analysis investigating the association between smoking and the development of bronchiectasis in young adults. The study is nicely performed, with limitations addressed. I have a few minor comments;

Response. Thank you for your positive and encouraging comments. We have provided our point-by-point responses below for each of your concerns.

Specific comments

C1. It is stated that those diagnosed with bronchiectasis more than one year before the enrollment period were excluded. What about those diagnosed within a year of enrollment? If they were included, how were they handled in the survival analysis?

R1. Thank you for bringing this to our attention. We apologize for not addressing this in the original manuscript (page 5, lines 97–104). We have clarified the exclusion criteria in the Methods section of the revised manuscript as follows and modified Figure 1 accordingly:

“This study initially included 6,861,282 individuals, aged 20–39 years, who participated in a health screening exam between January 1, 2009 and December 31, 2012. We excluded the participants who had missing information for at least one variable among those analyzed in this study (n = 543,778), those diagnosed with bronchiectasis before the enrollment period (n =35,005), those diagnosed with cystic fibrosis, situs inversus, or congenital bronchiectasis before the enrolment period (n = 376), those diagnosed with bronchiectasis within one year after enrollment (n = 6,560), and those who died within one year after enrollment (n = 6,498). Finally, a total of 6,275,575 participants were included in this study (Figure 1)."

C2. In model 3 comorbidities were adjusted for. Were all those diagnosed before the enrollment period? Were any diagnosed during follow-up, and if so, were they modelled as time-dependent covariates?

R2. We appreciate your insightful review of our manuscript. This study assessed the comorbidities of the patients during the enrollment period from January 1, 2009 to December 31, 2012 but not during the follow-up period. We have clarified this in the Methods section of the revised manuscript (page 7, line 143) as follows:

“Comorbidities were assessed during the enrollment period and…”

C3. The increased risk of bronchiectasis in ex-smokers is interesting. Were there any differences in this group based on their pack year history (greater or less than 10 pack years)? Was the risk seen with the higher pack-year exposure seen in current smokers also seen in the ex-smoker group?

R3. Thank you for your pertinent comments. Unfortunately, our current dataset lacks the pack-year history of ex-smokers. Because the entire NHIS dataset is very large, the NHIS only provides the data requested initially. As we had planned to analyze the pack-years for smokers, we did not request for data regarding pack-years for ex-smokers. Since an additional request for data is not permitted by the NHIS, we have added this as a limitation of our study in the Discussion section of the revised manuscript (page 14, lines 288–291) as follows:

Third, as we solely focused on the smoking amount and the risk of bronchiectasis in current smokers, the association between smoking amount and the risk of bronchiectasis in ex-smokers should be investigated in future studies.”

C4. Was there any association with bronchiectasis and time since smoking ceased?

R4. Thank you for your giving us an opportunity to address this concern. Since the time of smoking cessation is not available in the NHIS database, we could not evaluate this issue.

Reviewer 3 Report

This study aimed to detect whether smoking is a risk factor for incident bronchiectasis, which is indeed an important clinical question.

However, the study design used cannot answer this question in my opinion, since it is only based on diagnosis codes from clinic visits in the young general population. 

As well known, the diagnosis of bronchiectasis is only performed by chest imaging, specifically HRCT, which is rarely performed in healthy young adults (even smokers at this age, which do not qualify for annual lung cancer screening LDCT).   

Therefore, in most cases, unless performed for a different indication, a patient would undergo chest CT only in cases of chronic respiratory symptoms (which are obviously more common in smokers). More CT's would inarguably mean more diagnoses of bronchiectasis which are often an incidental finding. Therby refferal bias is a major flaw in this study design.

I believe that a better study design would compare CT scans of smokers with CT scans of age, gender and geographical region matched non-smokers. If bronchiectasis is indeed more frequent in the smoking group than that would indicate an association (although not proove causality).  

Author Response

## Response to Reviewer 3

General comment

This study aimed to detect whether smoking is a risk factor for incident bronchiectasis, which is indeed an important clinical question. However, the study design used cannot answer this question in my opinion, since it is only based on diagnosis codes from clinic visits in the young general population. As well known, the diagnosis of bronchiectasis is only performed by chest imaging, specifically HRCT, which is rarely performed in healthy young adults (even smokers at this age, which do not qualify for annual lung cancer screening LDCT). Therefore, in most cases, unless performed for a different indication, a patient would undergo chest CT only in cases of chronic respiratory symptoms (which are obviously more common in smokers). More CT's would inarguably mean more diagnoses of bronchiectasis which are often an incidental finding. Therby refferal bias is a major flaw in this study design. I believe that a better study design would compare CT scans of smokers with CT scans of age, gender and geographical region matched non-smokers. If bronchiectasis is indeed more frequent in the smoking group than that would indicate an association (although not proove causality).

Response. We appreciate your detailed review of our study and agree with these valuable comments. We defined bronchiectasis using the ICD-10 diagnosis code rather than chest computed tomography. Hence, this study may have misclassification bias, which is also a major limitation of population-based studies. Nonetheless, we believe that this study has the value of the generation hypothesis, which warrants further studies to confirm our findings. Accordingly, we have acknowledged this as a study limitation in the Discussion section of the revised manuscript (page 14, lines 284–286) as follows:

, the diagnosis of bronchiectasis was based on the ICD-10 diagnosis code and not on chest computed tomography results. Thus, there could be misclassification of the diagnosis of bronchiectasis.”

Round 2

Reviewer 3 Report

Refferal and detection bias are not only limitations of this study, but in my opinion preclude any conclusion regarding smoking being a risk factor for bronchiectasis. The data only suggests that smokers may be more prone to be diagnosed with bronchiectasis. 

Author Response

## Response to reviewer 3

Comments. Refferal and detection bias are not only limitations of this study, but in my opinion preclude any conclusion regarding smoking being a risk factor for bronchiectasis. The data only suggests that smokers may be more prone to be diagnosed with bronchiectasis.

Response. Thank you for your comments. We agree with the reviewer that our study only suggests that smokers may be more prone to be diagnosed with bronchiectasis. We toned down that we showed the association of smoking with bronchiectasis in the revised manuscript.
